# Mammalian Orthoreovirus (MRV) Is Widespread in Wild Ungulates of Northern Italy

**DOI:** 10.3390/v13020238

**Published:** 2021-02-03

**Authors:** Sara Arnaboldi, Francesco Righi, Virginia Filipello, Tiziana Trogu, Davide Lelli, Alessandro Bianchi, Silvia Bonardi, Enrico Pavoni, Barbara Bertasi, Antonio Lavazza

**Affiliations:** 1Istituto Zooprofilattico Sperimentale della Lombardia e dell’Emilia Romagna (IZSLER), 25124 Brescia, Italy; sara.arnaboldi@izsler.it (S.A.); francesco.righi@izsler.it (F.R.); tiziana.trogu@izsler.it (T.T.); davide.lelli@izsler.it (D.L.); enrico.pavoni@izsler.it (E.P.); barbara.bertasi@izsler.it (B.B.); antonio.lavazza@izsler.it (A.L.); 2National Reference Centre for Emerging Risks in Food Safety (CRESA), Istituto Zooprofilattico Sperimentale della Lombardia e dell’Emilia Romagna (IZSLER), 20133 Milan, Italy; 3Istituto Zooprofilattico Sperimentale della Lombardia e dell’Emilia Romagna (IZSLER), 23100 Sondrio, Italy; alessandro.bianchi@izsler.it; 4Veterinary Science Department, Università degli Studi di Parma, 43100 Parma, Italy; silvia.bonardi@unipr.it

**Keywords:** mammalian orthoreovirus, wild ungulates, reservoir, wildlife

## Abstract

Mammalian orthoreoviruses (MRVs) are emerging infectious agents that may affect wild animals. MRVs are usually associated with asymptomatic or mild respiratory and enteric infections. However, severe clinical manifestations have been occasionally reported in human and animal hosts. An insight into their circulation is essential to minimize the risk of diffusion to farmed animals and possibly to humans. The aim of this study was to investigate the presence of likely zoonotic MRVs in wild ungulates. Liver samples were collected from wild boar, red deer, roe deer, and chamois. Samples originated from two areas (Sondrio and Parma provinces) in Northern Italy with different environmental characteristics. MRV detection was carried out by PCR; confirmation by sequencing and typing for MRV type 3, which has been frequently associated with disease in pigs, were carried out for positive samples. MRV prevalence was as high as 45.3% in wild boars and 40.6% in red deer in the Sondrio area, with lower prevalence in the Parma area (15.4% in wild boars). Our findings shed light on MRV occurrence and distribution in some wild species and posed the issue of their possible role as reservoir.

## 1. Introduction

Reoviruses belong to the *Reoviridae* family (the term “reo” is an acronym for “respiratory enteric orphan” viruses) [1], which comprises viruses infecting humans, animals, plants, and insects [2]. Reoviruses, which preferentially infect mammals, birds, and reptiles, are grouped in the genus *Orthoreovirus* [3]. These non-enveloped viruses of 60–100 nm in diameter have several common traits, including a double-layer icosahedral capsid and a genome composed of 10 fragments of double-stranded RNA (dsRNA) coding for 12–13 proteins. The genome total size is approximately 23.500 bp [4,5]. Mammalian orthoreovirus (MRV) was the first *Orthoreovirus* species isolated from humans in 1950, and it has been the major model system for molecular understanding of the *Reoviridae* family [6,7,8]. 

Three MRV serotypes have been described, being differentiated by the capacity of anti-reovirus sera to neutralize viral infectivity and inhibit hemagglutination [6]. Each serotype is represented by a prototype strain isolated from a human host, therefore are designated as type 1 Lang, type 2 Jones, and type 3 Dearing (T1L, T2J, and T3D, respectively).

MRV dsRNA segments are classified as large (L1, L2, and L3), medium (M1, M2, and M3), and small (S1, S2, S3, and S4) based on their size and electrophoretic mobility [9], for a total genomic size of 23.606 bp for T1L, 23.578 bp for T2J, and 23.560 bp for T3D [2]. MRVs are prone to genetic reassortment and intragenic rearrangements due to the segmented nature of their genome that can lead to evolution dynamics with unpredictable biological properties and a potential host range expansion [2,10,11,12]. Because of the apparent lack of species barriers, MRVs can infect humans and a broad range of mammalian species including livestock, companion, and wild animals [11,13,14,15]. Infection was documented in farmed white-tailed deer [10], bats [7,16,17] and chamois [11]. MRVs can cause respiratory, central nervous system, and enteric diseases both in human [13,18], and in other mammalian hosts [7,15,19]. Indeed, neurological clinical signs in baboons and snakes, and enteritis and pneumonia in swine have been described [3]. MRV zoonotic transmission has been reported due to direct contact as well as indirect transmission [13]. MRVs are relatively stable outside the host and, therefore, they can efficiently persist in the environment. Indeed, reoviruses are frequently detected in a wide variety of environmental sources like surface water, seawater, and wastewater [20,21,22,23]. Therefore, awareness of MRVs as new emerging infectious agents in wild animals is essential to minimize the risk of its spreading to farm animals and possibly to humans.

MRV type 3 (MRV3) was proven to be pathogenic for pigs causing enteritis, pneumonia, and encephalitis [15,19]. Recently, it has been associated with severe diarrhea and respiratory clinical signs also in China, Korea, and the United States [3,15,24,25]. MRV3 may play a role in outbreaks of acute gastroenteritis in piglets, also co-infecting with other enteric pathogens, such as porcine epidemic diarrhea virus (PEDV) and porcine deltacoronavirus (PDCoV) [3,15,24]. Additionally, neurovirulent MRV3 strains have been isolated from dogs with diarrhea in Japan [26], and in Italy [27]. Nevertheless, the pathogenic potential of MRV3 is still unclear and specific data concerning the virus ecology, transmission routes, and the possible role of wild fauna as MRV reservoir are lacking.

Therefore, the aim of this study was to investigate the occurrence and distribution of likely zoonotic MRVs in wild ungulates in Northern Italy to assess the potential role of these animals as reservoirs.

## 2. Materials and Methods

### 2.1. Sampling

This study was carried out within a monitoring plan for Hepatitis E virus (HEV) in game animals, including wild boars, red deer, roe deer, and chamois. Liver samples from wild boars (*Sus scrofa*), were collected between 2018 and 2020 during hunting seasons in two areas in Northern Italy characterized by different environmental features. One area near Parma, in the Po Valley (flatland), is highly anthropic and with a high pig farm density. In this territory a considerable wild boar population is present (about 12 animals/100 ha) and programmed depopulation plans are applied [28]. The other area was in the Sondrio province, and it is characterized by mountains, low anthropization of the territory, low farm density and high conservation of natural ecosystems.

Between 2016 and 2019 hunting seasons, liver samples were also collected from red deer (*Cervus elaphus*) in the Sondrio province. Moreover, during the 2018 hunting season, liver samples were also collected from chamois (*Rupicapra rupicapra*) and roe deer (*Capreolus capreolus*) settled in the Sondrio province. In this area, wild ruminants are widespread, and monitoring studies indicate densities of about 15 deer/100 ha, 5–10 roe deer/100 ha, and 3–10 chamois/100 ha [29].

### 2.2. Sample Preparation and RNA Extraction

For MRV detection, ca 50 g of tissue were taken from each liver. They were finely cut and 450 mg of each sample, were weighed and divided into three sterile 2 mL tubes each containing 150 mg of liver, 1.5 mL of QIAzol Lysis Reagent (Qiagen, Hilden, Germany) and 2 glass beads (5 mm diameter). Then, 10 μL of Mengovirus (recombinant Mengovirus—vMC0; 10^4^ viral particles/μL) were spiked into every sample as process control, as suggested by the UNI CEN ISO/TS 15216 2:2013). The samples were homogenized with the TissueLyser (Qiagen, Hilden, Germany) at 30 Hz for 10 min. After incubation at room temperature for 5 min, 0.2 mL of Chloroform (1 M) were added to each tube. Samples were left at room temperature for 5 min and after centrifugation at 7000× *g* for 3 min at 4 °C, the supernatant was ready for RNA purification. Viral RNA was purified using the NucliSENS^®^ MiniMag kit (bioMérieux SA, Marcy-l’Etoile, France) according to the manufacturer’s instructions. The eluted RNA was stored at −80 °C until use.

### 2.3. Reverse Transcription

Reverse Transcription (RT) was performed on a GeneAmp^®^ PCR System 9700 thermal cycler (Applied Biosystems, Foster City, CA, USA) in a total volume of 40 μL, containing 4.5 μL of viral RNA from the samples and 35.5 μL of reaction mixture. The reaction mixture contained 16 μL of dNTPs pool (10 mM), 8 μL of MgCl_2_ (25mM), 8 μL of Transcriptase Buffer (5X) (Invitrogen, Carlsbad, CA, USA), 2 μL of Random Hexamers (50 μM) (Invitrogen, Carlsbad, CA, USA), 1 μL of RNAse Inhibitor (40 U/μL) (Invitrogen, Carlsbad, CA, USA) and 0.5 μL of M-MLV Reverse Transcriptase (200 U/μL) (Promega Corporation, Madison, WI, USA). Each run included a negative control, which contained water in place of RNA, and a positive control (4.5 μL of MRV type 1 Lang–T1L RNA). RT was carried out at 42 °C for 60 min, and the reaction was heated at 94 °C for 5 min, then cDNA was available for nested-PCR.

### 2.4. Nested-PCR

MRV was detected by nested-PCR using specific primers described by Leary et al. [30], for the MRV L1 gene region, encoding for a component of the viral RNA-dependent RNA polymerase and conserved among reovirus strains [30]. Primers used for PCR amplification are shown in Table 1. For the primary PCR L1-rv5 and L1-rv6 primers were used and for the secondary PCR L1-rv7 and L1-rv8 primers were used.

The reaction was performed for both PCRs in a total volume of 25 µL containing 5 µL of Green GoTaq Flexi Buffer (5×) (Promega Corporation, Madison, WI, USA), 2 µL of MgCl_2_ (25 mM), 1 µL of dNTPs pool (10 mM), 0.25 µL of each primer (50 µM), 0.125 µL of Go Taq Flexi DNA polymerase (5 U/µL) (Promega Corporation, Madison, WI, USA), and 11.375 µL of DNAse-RNase-free water (Sigma–Aldrich, St. Louis, MO, USA), and 5 µL of cDNA template were added to the reaction mix.

Each run included a negative control, which contained water in place of cDNA, and a positive control (5 μL of MRV T1L cDNA of the RT positive control). Both PCRs were performed on a GeneAmp^®^ PCR System 9700 thermal cycler (Applied Biosystems, Foster City, CA, USA) with the following thermal profile: 5 min at 94 °C, then 40 cycles of 30 s at 94 °C, 1 min at 54 °C and 1 min at 72 °C, and a final incubation at 72 °C for 7 min.

Then, L1 gene PCR products were loaded into 2.5% agarose gel (Agarose MP, Roche, Basel, Switzerland), stained with EuroSafe Nucleic Acid Stain (EuroClone, Pero, Milan, Italy). A 100 bp marker (Invitrogen, Carlsbad, CA, USA) was used for electrophoretic run. Agarose gel was then displayed on the FireReader V10 transilluminator (UVITEC Cambridge, Rugby, UK) to observe the expected bands.

### 2.5. Mengovirus Real-Time PCR

Mengovirus cDNA detection was used to validate the process: if a sample was negative for Mengovirus the analysis was repeated from the sample preparation step. Real-time PCR was performed using primers and TaqMan probe shown in Table 2 to confirm the process effectiveness.

The reaction was performed using RNA UltraSense™ One-Step Quantitative RT-PCR System (Invitrogen, Carlsbad, CA, USA) in a total volume of 25 µL containing 5 µL of Ultrasense reaction mix (5×), 1 µL of each primer (12.5 µM and 22.5 µM, Forward and Reverse respectively), 1 µL of probe (6.25 µM), 0.5 µL of Rox reference dye (50×), 1.25 µL of RNA Ultrasense enzyme mix and 10.25 µL of DNAse-RNase-free water (Sigma–Aldrich, St. Louis, MO, USA). Five µL of RNA template were added to the reaction mix, and positivity was detected in each sample, highlighting data reliability.

The reaction was performed in a CFX96 Touch™ Real-Time PCR Detection System (Bio–Rad, Hercules, CA, USA). RT was performed for 1 h at 55 °C, and then samples were incubated at 95 °C for 5 min and amplified for 45 cycles of 15 s at 95 °C, 1 min at 60 °C and 1 min at 65 °C. Each analysis included a negative control which contained water in place of RNA.

### 2.6. MRV Type 3 Typing PCR

The samples positive for MRV nested-PCR were analyzed to detect MRV3 serotype using a booster PCR with specific primers as previously described [7,27], targeting the S1 gene. Primers used for PCR amplification are shown in Table 3.

The reaction was performed twice in a total volume of 25 µL containing 5 µL of Green GoTaq Flexi Buffer (5×), Promega Corporation, Madison, WI, USA), 2 µL of MgCl_2_ (25 mM), 1 µL of dNTPs pool (10 mM), 0.25 µL of each primer (50 µM), 0.125 µL of Go Taq Flexi DNA polymerase (5 U/µL, Promega Corporation, Madison, WI, USA) and 11.375 µL of DNAse-RNase-free water (Sigma–Aldrich, St. Louis, Missouri, USA). Five µL of cDNA template or amplified were added to the reaction mix. Each run included a negative control which contained water in place of DNA, and a positive control (5 μL of MRV3 cDNA template or previously amplified).

PCR was performed on a GeneAmp^®^ PCR System 9700 thermal cycler (Applied Biosystems, Foster City, California, USA) with the following thermal profile: 1 min at 94 °C, then 35 cycles of 20 s at 94 °C, 30 s at 50 °C and 30 s at 72 °C, and a final incubation at 72 °C for 10 min.

PCR products were loaded into 2.5% agarose gel (Agarose MP, Roche, Basel, Switzerland), stained with EuroSafe Nucleic Acid Stain (EuroClone, Pero, Milan, Italy). A 100 bp marker (Invitrogen, Carlsbad, California, USA) was used for electrophoretic run. Agarose gel was then displayed on the FireReader V10 transilluminator (UVITEC Cambridge, Rugby, UK) to observe expected bands.

### 2.7. Sequencing

The samples positive for MRV nested-PCR were sequenced to confirm virus identification. The positive amplification products were first enzymatically purified using FastAP™ Thermosensitive Alkaline Phosphatase (1 U/µL) and ExonucleaseI (20 U/µL) (Thermo Fisher Scientific, Waltham, MA, USA) according to the manufacturer’s instructions. The forward and reverse sequence reactions were prepared separately using the L1 primers listed in Table 1 on a GeneAmp^®^ PCR System 9700 thermal cycler (Applied Biosystems, Foster City, CA, USA). Each reaction was performed in a total volume of 10 µL containing 2 µL of Big Dye Terminator Reaction Mix, 1 µL of Big Dye Terminator 5× Sequencing Buffer, 3 µL of DNAse-RNase-free water (Sigma–Aldrich, St. Louis, Missouri, USA), and 2 µL of primer (1.6 µM); finally, 2 µL of purified amplification product were added in each reaction tube. Purified products were incubated at 96 °C for 1 min and then amplified for 25 cycles of 96 °C for 10 s, 50 °C for 5 s, and 60 °C for 4 min.

Sequence reaction products were purified using the BigDye XTerminator^®^ Purification Kit (Thermo Fisher Scientific, Waltham, MA, USA), according to the manufacturer’s instructions. Samples were finally sequenced by the SeqStudio Genetic Analyzer (Applied Biosystem Inc., Foster City, CA, USA).

The consensus sequences were created and aligned with the MEGA-6 software [31]. The generated sequences were compared for similarity against all virus sequences deposited in the NCBI GenBank database using the Basic Local Alignment Search Tool (BLAST) to confirm the PCR result. The nucleotide sequences of 20 MRV L1 fragments have been deposited in the GenBank database (Accession Numbers: wild boars from MW191815 to MW191824, red deer from MW191825 to MW191834).

### 2.8. Viral Isolation

Among PCR positive samples a selection was made based on animal species, year, and area of origin (Appendix A). The selected liver samples (*n* = 11) were homogenized in minimal essential medium (1 g/10 mL) containing antibiotics and clarified by centrifugation at 3000× *g* for 15 min. Supernatants were inoculated in 24 well plates containing confluent monolayers of VERO (African green monkey) and MARC-145 (Fetal monkey) kidney cells, incubated at 37 °C with 5% CO_2_ and observed daily for 7 days to detect the development of cytopathic effect (CPE). In the absence of CPE, the cryolysates were sub-cultured twice onto fresh monolayers, before being discharged as negative.

### 2.9. Statistical Analysis

The prevalence was calculated by animal species, considering all animals tested (overall), and for each descriptive variable included in the three models, with the 95% CI and the *p*-value for statistical significance. Through logistic regression, it was tested whether MRV infection was influenced by collection year, sampling area, age class, gender, or pregnancy status of the wild animals sampled (independent variables) and considered the outcome as the dependent variable. In particular, the “wild boar” model included all wild boars analyzed (*n* = 336) and considered all variables, the “wild ruminant” model comprised red deer, chamois, and roe deer (*n* = 128), and considered year of sampling, age class, and gender as independent variables. Finally, the “overall” model included all animals (*n* = 464), and since data on ruminants were collected from 2016 to 2018 and only in the Sondrio province (while those on wild boars were collected starting from 2018 in both areas), the collection year and sampling area variables were not included in this model.

Moreover, among the positives, ORs were calculated to verify the association between the MRV3 and sampling area or animal species.

All statistical elaborations were carried out using R statistical software (3.6.2. version), and the 5% statistical significance. MRV prevalence, and the 95% confidence interval (95% CI) were calculated with the Blaker’s method.

## 3. Results

### 3.1. Sampling and MRV Prevalence

A total of 336 liver samples were collected from wild boars of both genders divided in three age classes: class 0—less than 1 year-old animals, class 1—between 1 and 2 years old and class 2—more than 2 years old; young, sub-adult, and adult animals, respectively (Table 4). From red deer, 118 liver samples were collected in the Sondrio province (Table 5). Finally, 10 liver samples were also collected from 6 chamois, and 4 roe deer settled in the Sondrio province.

A total of 150 samples (*n* = 100 wild boars, and *n* = 50 wild ruminants) were positive for MRV, with an overall prevalence in wild ungulates of 32.3% (95% CI 28.2–36.7%) as reported in Table 6.

MRV prevalence was higher in wild ruminants (39.0%, 95% CI 31.0–47.7%) vs. wild boars (29.7%, 95% CI 25.1–34.8%) (*p* = 0.028). However, the prevalence was not statistically different considering gender and age. In fact, an overall lower prevalence was observed in adults (class 2) when compared to class 0 (*p* = 0.078), or to sub-adults (class 1, *p* = 0.064), with *p*-values closed to significance. All OR values, related to age and gender, calculated by logistic models were not significant except OR related to collection year in wild boars (*p* = 0.037).

Concerning wild boars, 100 liver samples were positive for MRV (29.7%; 95% CI 25.1–34.8%), with an overall prevalence of 45.3% (95% CI 37.8–53.0%) and 15.4% (95% CI 10.8–21.5%), in Sondrio and Parma areas, respectively (Table 7). In the Sondrio province, a statistically significant higher positivity than in the Parma area was observed (*p* < 0.0001). Indeed, MRV was detected in 73 out of 161 (45.3%) liver samples in the Sondrio province, and in 27 out of 175 (15.4%) liver samples in the Parma area. Moreover, in 2019–2020, a higher prevalence was reported (*p* = 0.003). Certainly, an increasing trend from 2018 to 2020 was observed (30.4% 2018–2019, and 60.7% 2019–2020) in the Sondrio province, while prevalence in wild boar was similar during the two hunting seasons (15.2% 2018–2019, 15.7% 2019–2020) in the Parma area. Furthermore, an OR = 0.50 (95% CI 0.10–1.95, *p* = 0.35) associated with pregnancy status was calculated, showing that pregnancy did not represent a risk factor for MRV infection.

Concerning wild ruminants, in red deer MRV prevalence was as high as 40.6% (95% CI 32.2–49.7%); MRV was also detected in 2 out of 6 chamois, but in none of the roe deer. From 2016 to 2019 the prevalence first increased to 65.6% (2017–2018, *p* = 0.049) and then decreased down to 8.3%, (*p* = 0.001) in 2018–2019.

### 3.2. MRV Type 3 Typing PCR

In total, 24 out of 100 (24.0%; 95% CI 16.6–33.2%) positive wild boar samples were typed as MRV3. Among MRV positive samples, MRV3 represented 21.9% and 29.6% of the samples collected in the Sondrio and Parma provinces, respectively. Considering wild ruminants, five out of 48 (10.4%; 95% CI 4.5–22.1%) red deer positive samples were typed as MRV3, while all chamois samples were negative for this serotype. Moreover, MRV3 serotype did not show any significant correlation with the animal species or sampling area.

### 3.3. Confirmation by Sequencing

A total of 21 wild boar and 24 deer samples were successfully sequenced, showing a 100% identity value with the MRV type 1, 2, and 3, L1 genes.

### 3.4. Viral Isolation

MRV isolation was not successful. Indeed, none of the 11 liver samples submitted to viral isolation caused any apparent CPE after three blind passages.

## 4. Discussion

Wild animals are increasingly recognized as important reservoirs for viruses of proven or potential significance for human and veterinary health. To investigate their role as MRV reservoir, animals living in two areas in Northern Italy were compared. In our study, MRV prevalence in wild boars was as high as 45.3% (95% CI 37.8–53.0%) in the Sondrio province, an area characterized by mountains. In this area, such high prevalence raises minor concerns due to both low anthropic activity and low farm density. In the Parma area, MRV prevalence in wild boars was lower (15.4%; 95% CI 10.8–21.5%), and such observation seems reassuring in consideration of the high density of pig farms, where the animals are raised for one of the greatest Italian excellences, the Parma Ham. The two investigated areas have a different wild boar population density, with Parma characterized by a higher density (12 animals/ha) and Sondrio by a lower one. In this area, the actual size of the population is unknown since wild boar are considered allochthonous and therefore subjected to eradication, as their presence in the mountain area is attributable to illegal repopulation and not to natural settlement. Our results suggest that, however, such low population density did not hamper the virus transmission, hinting that other risk factors may play a role. Indeed, the ability of MRV to persist outside the host may contribute to its diffusion and it is an advantageous feature that probably enables viral transmission also in less populated areas [20,21,22,23]. Furthermore, since wild boars are omnivorous scavengers, the overlap in the use of space for food purposes by different species, including wild ruminants, could allow environmental contamination, which can lead to an increased viral circulation [32].

Regarding the possible risk factors associated with the infection in wild boars, no statistically significant differences in relation to gender or age class were found. Nevertheless, we observed an overall MRV prevalence decrease from young animals to adults. This is consistent with studies on mice reporting a strong correlation with host age, with neonatal mice more susceptible to viral infection [33,34]. Moreover, despite MRV manifestations being rare in humans, also in humans MRV infections are more common in early childhood following the loss of maternal antibodies, with seroprevalence increasing from 8 to 50% from 1 to 5 years of age [13,35]. MRVs were recently isolated in Japan and in Europe (Slovenia and Switzerland) from children with meningitis, acute gastroenteritis, or diarrhea, and a zoonotic transmission was suspected, probably caused by the contact with domestic or farm animals and animal feces [13,36,37].

At present, data on MRV infection in wild ruminants are scarce. A study on white-tailed deer in the United States suggested MRV reassortment from other animal species [10], and a study on alpine chamois reported a high seroprevalence in Italy, suggesting an endemic level of infection in this population [11]. In the Sondrio province, wild ruminants are widespread [29], and monitoring studies indicate higher densities compared to the Parma flatland in which wild ruminants are seldom spotted. In our study, a higher MRV prevalence was found in wild ruminants (39.0%) compared to wild boars (29.7%), suggesting their likely role as MRV reservoirs. This may suggest a higher risk of MRV infection for wild boars settled in the Sondrio province posed by the shared habitat and due to interspecies transmission. 

In our study, MRV3 was identified in 24.0% of the wild boar samples (21.9% and 29.6% of the positive samples in the Sondrio and Parma areas, respectively), and in 10.4% of the red deer samples. This serotype is a proven pig pathogen, and it has been frequently associated with other enteric viruses, (e.g., PEDV, PDCoV) both in the United States and in Italy in 2015 [15,19].

Regarding viral isolation, among the possible hypotheses that can explain the failure of isolation on cell cultures are low initial viral titer, different matrix used as inoculum (liver vs. intestine), poor sample quality, inadequate storage, or repeated freezing/thawing with consequent loss of viability of the virus.

To the best of our knowledge, this is the first study on MRV prevalence in wild ungulates. Our results contribute to extend the current knowledge on the role of wild boar and red deer as potential MRV reservoirs and emphasize the importance of monitoring efforts in wild animals, and in other likely reservoirs.

To further investigate MRV transmission dynamics and epidemiology, phylogenetic studies on viral strains collected from different hosts are essential. It will also be very important to investigate the role of roe deer in the MRV transmission, as this species is widely spread in the two study areas and shares the use of the habitat with the wild boar. Since MRV reassortment ability likely contributes to its genetic evolution and adaptation to different mammalian hosts, such studies might contribute to understanding the zoonotic potential and pathogenesis of different viral clusters.

## 5. Conclusions

In this study, a higher MRV prevalence in the wild boar population of a mountainous territory (Sondrio, 45.3%) compared to a lower prevalence in a flat area (Parma, 15.4%) was observed despite the higher farm density that characterizes Parma municipality, which might suggest an easier viral circulation. Moreover, MRV appears to be widely distributed among wild animals in the Sondrio province, with prevalence in wild ruminants as high as 65.6%. Our findings shed light on some aspects of occurrence and distribution of MRV in wild ungulates and posed the issue of wild boar and deer as possible reservoirs. Enhancing the awareness of MRV as a new emerging infectious agent that may affect wild animals is essential to minimize the risk of it spreading to farm animals and, possibly, to humans.

## Figures and Tables

**Table 1 viruses-13-00238-t001:** Primers used for PCR amplification.

Name	Type	Sequence	Position ^1^	Amplicon Size
*L1-rv5*	Forward	5′–GCATCCATTGTAAATGACGAGTCTG–3′	1888–1912	416 bp
*L1-rv6*	Reverse	5′–CTTGAGATTAGCTCTAGCATCTTCTG–3′	2278–2303	
*L1-rv7*	Forward	5′–GCTAGGCCGATATCGGGAATGCAG–3′	1930–1953	344 bp
*L1-rv8*	Reverse	5′–GTCTCACTATTCACCTTACCAGCAG–3′	2249–2273	

^1^ Primer position is referred to the L1 sequence of T1L/53 (accession: NC00_4271) [30].

**Table 2 viruses-13-00238-t002:** Primers and probe used to detect Mengovirus in Real-time PCR.

Name	Type	Sequence
Mengo 110	Forward	5′–GCGGGTCCTGCCGAAAGT–3′
Mengo 209	Reverse	5′–GAAGTAACATATAGACAGACGCACAC–3′
Mengo 147	Probe	5′–FAM–ATCACATTACTGGCCGAAGC–MGB–3′

**Table 3 viruses-13-00238-t003:** Primers used for PCR amplification.

Name	Type	Sequence	Position ^1^	Amplicon Size
*S1-R3F*	Forward	5′–TGGGACAACTTGAGACAGGA–3′	338–357	326 bp
*S1-R3R*	Reverse	5′–CTGAAGTCCACCRTTTTGWA–3′	644–663	

^1^ Primer position is referred to S1 MRV3 segment [7].

**Table 4 viruses-13-00238-t004:** Characteristics of the wild boars sampled in the two investigated areas (Sondrio–SO, and Parma–PR). The number of pregnant females is reported in brackets.

Sampling Area	Gender	Collection Year	Age Class ^1^			Total Samples
			0	1	2	
**SO**	**Male**	2018/2019	10	23	15	106
		2019/2020	12	34	12	
	**Female**	2018/2019	3	19	12	55
		2019/2020	5	11	5	
	**Total samples**		30	87	44	161
**PR**	**Male**	2018/2019	1	17	20	74
		2019/2020	13	6	17	
	**Female**	2018/2019	1	23 (6)	43 (32)	101 (38)
		2019/2020	6	9	19	
	**Total samples**		21	55	99	175

^1^ Class 0: young; class 1: sub-adults; class 2: adults.

**Table 5 viruses-13-00238-t005:** Characteristics of the red deer sampled divided by gender and age class.

Gender	Total Samples	Age Class ^1^		
		0	1	2
**Male**	48	19	14	15
**Female**	70	16	10	44
**Total samples**	118	35	24	59

^1^ Class 0: young; class 1: sub-adults; class 2: adults.

**Table 6 viruses-13-00238-t006:** MRV prevalence according to species, gender, age, collection year, and sampling area. Wild ruminants’ column accounts for red deer, roe deer, and chamois samples. Age class 0 refers to <1 year-old animals, age class 1 to animals between 1 and 2 years old, and age class 2 to animals >2 years old. In brackets 95% CI.

	Wild Ruminants (*n* = 128)	Wild Boars (*n* = 336)	Overall (*n* = 464)
**Gender**			
Males	39.8% (27.5–53.5%)	28.2% (21.4–36.1%)	38.2% (31.6–45.4%)
Females	40.3% (28.8–53.0%)	28.9% (21.2–38.1%)	33.1% (26.7–40.2%)
**Age class**			
Class 0	41.7% (26.9–58.1%)	29.8% (18.6–44.0%)	39.2% (29.4–50.0%)
Class 1	42.3% (25.2–61.5%)	30.4% (22.8–39.3%)	39.9% (31.9–48.5%)
Class 2	36.2% (25.1–49.0%)	25.6% (18.4–34.4%)	28.3% (22.3–35.2%)
**Collection year**			
2016–2017	43.3% (31.5–56.0%)		–
2017–2018	65.6% (47.9–79.8%) *		
2018–2019	8.3% (2.7–22.9%) *	21.2% (15.3–28.7%)	
2019–2020	–	37.2% (29.1–46.1%) *	
**Sampling area**			
Parma		15.4% (10.8–21.5%)	
Sondrio		45.3% (37.8–53.0%) *	

* *p* < 0.05.

**Table 7 viruses-13-00238-t007:** Data between 2018 and 2020 hunting seasons in the two investigated areas (Sondrio–SO, and Parma–PR). In brackets 95% CI.

Sampling Area	Collection Year	Number of Analyzed Samples	Number of MRV Positive Samples	MRV Prevalence (95% CI)
**SO**	2018–2019	82	25	30.4% (21.5–41.1%)
	2019–2020	79	48	60.7% (49.7–70.7%)
	Total	161	73	45.3% (37.8–53.0%)
**PR**	2018–2019	105	16	15.2% (9.6–23.3%)
	2019–2020	70	11	15.7% (9.0–25.9%)
	Total	175	27	15.4% (10.8–21.5%)

## Data Availability

The data presented in this study are available within the study itself and in the Appendix A attached to this paper.

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
