# Peer review of "Mammalian Orthoreovirus (MRV) Is Widespread in Wild Ungulates of Northern Italy"

_viruses, 2021, doi:10.3390/v13020238_

Round 1

Reviewer 1 Report

This manuscript describes virologic detection of mammalian reovirus infections among wild boar, deer and other animals in two regions of Italy. The study is of potential interest for viral prevalence/detection considerations and is fairly straightforward. Several recommendations will lead to a better description of this study:

The study set-up detects active MRV infection and thus may underestimate overall infection rates. Indeed, similarly to human infection, infection appears to be higher in younger animals. Some additional clarification and comparison to human epidemiology would strengthen the discussion.

The use of random hexamers and primers designed against conserved pan-RdRp sequences is a good strategy to capture as many MRV types as possible. However, the apparent focus on MRV3, combined with MRV1-specific sequencing is confusing. Please provide justification for focusing on MRV3 despite also providing sequence information that is 100% identical to MRV1. Elaborate.

Minor points:

Line 45: What is evidence for “high” mutation rate, and what is “high”?

Line 62-63: also in pigs?

Line 101: spun; convert rpm to xg

Line 314-315: But if MRV can infect a wide range of animals, the “low wild boar population density” may be irrelevant. This is partially addressed later in the paragraph, but please expand.

Author Response

Reviewer 1: This manuscript describes virologic detection of mammalian reovirus infections among wild boar, deer and other animals in two regions of Italy. The study is of potential interest for viral prevalence/detection considerations and is fairly straightforward. Several recommendations will lead to a better description of this study:

The study set-up detects active MRV infection and thus may underestimate overall infection rates. Indeed, similarly to human infection, infection appears to be higher in younger animals. Some additional clarification and comparison to human epidemiology would strengthen the discussion.

We thank the reviewer for the suggestion, unfortunately human epidemiologic data at present are scarce; we however added references to some studies reporting MRV infection in children and mice and discussed them in relation to our findings (see lines 290-298).

The use of random hexamers and primers designed against conserved pan-RdRp sequences is a good strategy to capture as many MRV types as possible. However, the apparent focus on MRV3, combined with MRV1-specific sequencing is confusing. Please provide justification for focusing on MRV3 despite also providing sequence information that is 100% identical to MRV1. Elaborate

We thank the reviewer for highlighting a probable source of misunderstanding. Indeed, we looked for MRV3 since this correlates to more severe manifestations in pigs, as previously stated. The sequencing was instead carried out only to confirm the PCR result since some non-specific amplifications could occasionally occur. Indeed, the pan-RdRp is common to all MRVs and 100% identity was found with sequences from MRV types 1, 2, and 3. We therefore modified the text for clarity and to avoid further misunderstanding (see line 266).

Minor points:

Line 45: What is evidence for “high” mutation rate, and what is “high”?

The sentence was referring to the aptitude for genetic reassortment documented in MRVs. Indeed, the phrasing as pointed out by the referee could be misleading and was therefore modified for clarity (see lines 45-46).

Line 62-63: also in pigs?

MRV type 3 has a demonstrated pathogenic potential for pigs. The citation was modified for clarity (see lines 59-60).

Line 101: spun; convert rpm to xg

The text was modified also following the other reviewer comments (see line 92).

Line 314-315: But if MRV can infect a wide range of animals, the “low wild boar population density” may be irrelevant. This is partially addressed later in the paragraph, but please expand.

We understand the reviewer’s objection, and indeed, the consideration of low wild boar population density could be irrelevant, was the low species-specificity a certain fact. At present, the evidences to support such behaviour are still being collected. However, due to an extensive editing of the discussion the sentence has been removed (see lines 279-288).

Reviewer 2 Report

For the Authors: In this manuscript the authors (Arnaboldi et al.) present their studies of the occurrence of mammalian orthoreovirus (MRV) in wild ungulates (wild boars, two deer species and chamois) in two areas of Northern Italy in order to get a better understanding of reservoirs for zoonotic spread. This study appears to be a subproject to a (larger?) study of hepatitis E virus (HEV) in wildlife – and the authors refer in an important manner to unpublished data for HEV study in the discussion.  This is a notable drawback for understanding the overall public health implications of their study and it is highly recommended that the authors include the HEV-related results in this publication to put everything into a big-picture context rather than “slicing up” information in several publications (or hard-to-access internal or government reports).

In the Discussion the authors also mention a number of areas that should be done, including phylogenetic studies on viral strains (or isolates?) collected from different hosts. So why did the authors not do so?

Otherwise, the technical approaches appear overall appropriate, although it is disappointing that the authors were unable to isolate virus from the samples. Perhaps they should have gone beyond just two blind passages (minimum 5-7 is a more normal cut off) and/or tried other cell types?

The manuscript could be further improved by addressing the following:

  1. Line 14: change to “associated with asymptomatic….”
  2. Line 53: animals (and human babies) do not have “symptoms” but ‘clinical signs’. Symptoms is something the patient/sufferer can describe, while clinical signs is something an observer can register. Animals and human babies cannot describe how they feel, but the veterinarian or medical doctor/nurse can note clinical signs/changes in the subject. This comment also applies line 63: pigs do not have ‘symptoms’.
  3. Lines 59-60: here and in several other places (e.g., lines 277-78, 291-2, 294-5, 297-8, 344-7) in the manuscript there are one-sentence sections. That is inappropriate – so either include the sentence in the previous section or in the following. Whichever is more appropriate.
  4. Line 61: change to “pathogenic for pigs”.
  5. Line 101: change to “after centrifugation at 7000…..”
  6. 110: changed to “mixture contained ….”
  7. Line 155: change to “Five µL of RNA …..”. Never start a sentence with a numerical.
  8. Line 158: change to “performed for 1 hour”
  9. Line 177: change to “1 minute”
  10. Line 214: change to “plates containing confluent ….”
  11. Line 216: replace ‘highlight’ with “detect”
  12. Table 6: the column for wild ruminants is very confusing – and in general this table is very hard to interpret. Please modify to make it more clear and accessible to readers.
  13. Line 298: change to” viral isolation caused any apparent CPE …….”.
  14. Lines 317-8: change to “In contrast, our results …..”
  15. Lines 345: what do the authors mean by “it is possible to mention”? Lots of things are possible to mention, but maybe the authors mean “including”?

END

Author Response

In this manuscript the authors (Arnaboldi et al.) present their studies of the occurrence of mammalian orthoreovirus (MRV) in wild ungulates (wild boars, two deer species and chamois) in two areas of Northern Italy in order to get a better understanding of reservoirs for zoonotic spread. This study appears to be a subproject to a (larger?) study of hepatitis E virus (HEV) in wildlife – and the authors refer in an important manner to unpublished data for HEV study in the discussion.  This is a notable drawback for understanding the overall public health implications of their study and it is highly recommended that the authors include the HEV-related results in this publication to put everything into a big-picture context rather than “slicing up” information in several publications (or hard-to-access internal or government reports).

This study is part of a larger project carried out on MRVs aimed at identifying the presence of MRV in a range of animal species. MRVs can be zoonotic, and therefore it is of interest how MRVs are distributed across the different species. To date, we have already confirmed in previous studies the presence of MRV in our territory in bats, pigs, and chamois. Usually, this viral agent is investigated on samples collected in the framework of passive surveillance regional monitoring plans on wild fauna. The target organ for MRV is commonly the gastrointestinal tract, however testing the liver is an ideal indicator of how systemic the infection could be. In this case, we exploited an ongoing active monitoring plan for HEV that foresees liver collection from hunted animals, to investigate such aspect. It is furthermost from our intentions to voluntarily split the publications. The work on HEV is indeed focused on implications for food safety, while MRV has impacts for animal health and the zoonotic potential is still under evaluation. Therefore, conjoining results from these studies in a single paper would be to our opinion strained. To avoid further misunderstanding we decided to modify the discussion and avoid any reference to the HEV project.

In the Discussion the authors also mention a number of areas that should be done, including phylogenetic studies on viral strains (or isolates?) collected from different hosts. So why did the authors not do so?

The aim of this study was to identify species that could harbour MRVs and the extension of their diffusion. Indeed, it is our wish for the future to expand the range of species tested with dedicated active surveillance plans. For this study we focused on hunted fauna. Performing a phylogenetic analysis was not initially planned, and right now, to implement such analysis is not compatible with the current deadline for resubmission. We notified the issue to the Academic Editor, should her/him judge this analysis essential for publication, thus granting an extension of the deadline, we could try to perform sequencing of a part of the viral genome of some strains to allow the phylogenetic analysis.  

Otherwise, the technical approaches appear overall appropriate, although it is disappointing that the authors were unable to isolate virus from the samples. Perhaps they should have gone beyond just two blind passages (minimum 5-7 is a more normal cut off) and/or tried other cell types?

We were also disappointed to have not succeeded in isolating the MRV. This especially considering that in the previous studies on bats, pigs and chamois, we have always been able to isolate the virus quite easily. Indeed, we adopted exactly the same protocol by using two different permissive cell lines and then discharging as negative after three passages (7 days of observation each) with no CPE. However, differently from previous studies, here we tested the liver and not the intestine that is likely the main target organ. Therefore, we guessed that the lack of in vitro isolation could be due both to a very low viral load in such non target organ and likely to a not proper conservation condition. In fact, many samples were dated (since 2016) and in addition, as often occurs for samples coming from passive surveillance from hunting activity, they have been conserved at just -20°C and sometimes subjected to repeated freezing/thawing. Both these conditions are certainly inadequate to maintain viral vitality.

The manuscript could be further improved by addressing the following:

Line 14: change to “associated with asymptomatic….”

 The text was modified accordingly (see line 15).

Line 53: animals (and human babies) do not have “symptoms” but ‘clinical signs’. Symptoms is something the patient/sufferer can describe, while clinical signs is something an observer can register. Animals and human babies cannot describe how they feel, but the veterinarian or medical doctor/nurse can note clinical signs/changes in the subject. This comment also applies line 63: pigs do not have ‘symptoms’.

The text was modified accordingly (see lines 52 and 60-61).

Lines 59-60: here and in several other places (e.g., lines 277-78, 291-2, 294-5, 297-8, 344-7) in the manuscript there are one-sentence sections. That is inappropriate – so either include the sentence in the previous section or in the following. Whichever is more appropriate.

We thank the reviewer for pointing out this issue, the manuscript has been thoroughly checked and one-sentence sections have been included in previous or following (see lines 36, 45, 57, 67, 72, 93, 121, 132, 157, 171, 220-222, 242-250, 262, and 302).

Line 61: change to “pathogenic for pigs”.

The text was modified accordingly (see line 59).

Line 101: change to “after centrifugation at 7000…..”

The text was modified accordingly (see line 92).

Line 110: changed to “mixture contained ….”

The text was modified accordingly (see line 99).

Line 155: change to “Five µL of RNA …..”. Never start a sentence with a numerical.

The text was modified accordingly (see lines 139 and 156).

Line 158: change to “performed for 1 hour”

 The text was modified accordingly (see line 143).

Line 177: change to “1 minute”

The text was modified accordingly (see line 160).

Line 214: change to “plates containing confluent ….”

The text was modified accordingly (see line 193).

Line 216: replace ‘highlight’ with “detect”

The text was modified accordingly (see line 195).

Table 6: the column for wild ruminants is very confusing – and in general this table is very hard to interpret. Please modify to make it more clear and accessible to readers.

As suggested, Table 6 has been modified for clarity. In particular, the caption has been better detailed, the first row has been deleted to ease the reading, and age header has been simplified (see line 230-232 and Table 6).

 Line 298: change to” viral isolation caused any apparent CPE …….”.

The text was modified accordingly (see line 269).

Lines 317-8: change to “In contrast, our results …..”

 Due to an extensive modification to the Discussion section the term has been deleted.

Lines 345: what do the authors mean by “it is possible to mention”? Lots of things are possible to mention, but maybe the authors mean “including”?

As suggested, the sentence has been modified for style (see line 313-315).

Round 2

Reviewer 1 Report

My previous concerns have been addressed in the revised manuscript

Reviewer 2 Report

The authors are commended for addressing most of the points I raised in the first review-round. While the reasons given for not including the HEV-data in this publication are acknowledged and accepted, I still think it would have been better to place the results in a wider context, and that the impact of the study is diminished by only describing MRV occurrence.